# A simple method to measure methane emissions from indoor gas leaks

**Dominic Nicholas**[1]*, **Robert Ackley**[2], **Nathan G. Phillips**[3]

**1** HEET.org, Boston, MA, United States of America, **2** Gas Safety, Inc., Southborough, MA, United States of America, **3** Department of Earth and Environment, Boston University, Boston, MA, United States of America

* dominic.nicholas@heet.org

**Data Availability Statement:** All relevant data are within the manuscript and its Supporting Information files.

**Funding:** The author(s) received no specific funding for this work.

## Abstract

From wellhead to burner tip, each component of the natural gas process chain has come under increased scrutiny for the presence and magnitude of methane leaks, because of the large global warming potential of methane. Top-down measures of methane emissions in urban areas are significantly greater than bottom-up estimates. Recent research suggests this disparity might in part be explained by gas leaks from one of the least understood parts of the process chain: behind the gas meter in homes and buildings. However, little research has been performed in this area and few methods and data sets exist to measure or estimate them. We develop and test a simple and widely deployable closed chamber method that can be used for quantifying indoor methane emissions with an order-of-magnitude precision which allows for screening of indoor large volume ("super-emitting") leaks. We also perform test applications of the method finding indoor leaks in 90% of the 20 Greater Boston buildings studied and indoor methane emissions between 0.02–0.51 $ft^3$ $CH_4$ $day^{-1}$ (0.4–10.3 g $CH_4$ $day^{-1}$) with a mean of 0.14 $ft^3$ $CH_4$ $day^{-1}$ (2.8 g $CH_4$ $day^{-1}$). Our method provides a relatively simple way to scale up indoor methane emissions data collection. Increased data may reduce uncertainty in bottom-up inventories, and can be used to find super-emitting indoor emissions which may better explain the disparity between top-down and bottom-up post-meter emissions estimates.

## Introduction

Methane ($CH_4$) is the main component of natural gas (NG) and when leaked uncombusted into the atmosphere is a potent greenhouse gas having 86 times the global warming potential (GWP) of carbon dioxide over the first 20 years [1]. According to the Environmental Protection Agency (EPA) Inventory of U.S. Greenhouse Gas Emissions and Sinks (a national greenhouse gas inventory, or GHGI), methane makes up an estimated 11% of all greenhouse gasses, just over a third of which is attributed to NG and petroleum systems [2]. In urban areas with both high population and NG infrastructure density, studies show the NG distribution system to be one of the largest contributors of methane emissions [3–7]. Sources of methane emissions within the NG distribution system include metering and regulating stations, transmission pipelines, mains and service pipes, meters and NG appliances behind those meters [8].

**Competing interests:** The authors have declared that no competing interests exist.

Methane emissions can be estimated using top-down and bottom-up approaches. Top-down approaches model emissions from large areas or regions by using atmospheric data gathered from observation towers, aircraft and satellites. Bottom-up approaches measure or estimate methane emissions from collections of individual components in the system and typically involve multiplying the number of components (i.e. activity factors) by the emissions from those components (i.e. emission factors) which in turn can be determined using methods such as flux chambers and whole-building mass balance. A National Academy of Sciences report [9] highlights the need to improve the accuracy and precision of bottom-up methane emission estimates, focusing on areas of the inventory where there is the most uncertainty. The report suggests reasons for these uncertainties such as unaccounted-for sources, temporal and spatial variability of emissions, episodic high-emitting sources and inadequate spatial coverage of observational networks.

Recent studies have found that top-down methane emission inventories in urban areas are significantly greater than bottom-up methane emission inventories [3, 10, 11] and suggest that a potentially significant proportion of this disparity may be explained by post-meter NG methane emissions. Furthermore, studies have also correlated NG emissions with NG consumption [10, 12]. Whilst interest in post-meter emissions has been growing, few studies have been done to estimate or measure post-meter methane emissions. These studies have focused on measuring methane emissions from appliances such as domestic water heaters and stoves [13–15] and only one study in 2018 by Fischer et al. [16] measured quiescent whole-house methane emissions. Fischer et al studied 75 single-family owner-occupied California homes and appliances therein using a mass balance method implemented using commercial blower door systems and a portable gas analyzer. Many other buildings of different types, occupancy, age etc. need to be studied to be able to more accurately estimate post-meter emissions. In 2022 the EPA GHGI (a bottom-up methodology) was updated to include post-meter methane emissions for the first time and includes residential, commercial, industrial and power plants and NG vehicles sources. Residential post-meter emissions in this inventory were calculated retroactively for years 1990–2020 (estimates are not provided beyond 2020), using an emission factor based on the Fischer et al whole-house study. Residential post-meter methane emissions in 2020 accounted for 42% (192 kt $CH_4$) of all post-meter emissions. Similarly in 2022, the EPA GHGI post-meter inventory sources were added to the Massachusetts Department of Environmental Protection GHGI [17] also for years 1990–2020, with 2020 post-meter residential methane emissions accounting for 65% (4.5 kt $CH_4$) of all post-meter emissions as calculated using the EPA's emissions factor. These recent additions emphasize the recognition and importance of post-meter methane emissions inventories and present an opportunity to improve them.

The objective of our study is to develop a method to measure NG methane emissions into the indoors from NG infrastructure and appliances by approximating a room as a closed-system dynamic flux chamber [4, 18–20]. We develop air $CH_4$ concentration sampling approaches and flux calculation methods to arrive at a simple method that could be widely deployed. This scaling provides opportunities to improve our temporal and spatial understanding of indoor NG methane emissions across different building types (e.g. multi-family homes, businesses), across a diversity of socioeconomics and geography, and to further explore the super-emitting nature of indoor NG emissions [16, 21, 22].

## Materials and methods

### Measurements overview

We performed our study in basements with closable doors and windows to the exterior and upper floors of the building as rooms. Basements often contain a large amount of NG

infrastructure and appliances and thus where we may expect the majority of NG leakage to be occurring. We explored two methods to measure $CH_4$ emissions into basements: a high frequency intensive sampling room chamber method ("RCM") and a simplified room chamber method ("bag method"). We used a Picarro GasScouter G4301 Analyzer cavity ring-down spectrometer ("GasScouter") to collect air $CH_4$ concentration measurements. The GasScouter's inlet was connected to a tube with an intake at 127 cm from the ground positioning it approximately midway between floor and ceiling indoors. The GasScouter was periodically tested using test gasses (see S9 Appendix section 1). NG appliances were left in their normal operating states. We collected air $CH_4$ concentration measurements from outside of the building to determine an atmospheric $CH_4$ baseline, and from each floor of buildings, with basement rooms opened and measured last to minimize air mixing through basement doorways with upper floors due methane's buoyant nature. As a preliminary indication of measurable leakage, we tested the methods in basements having an average air $CH_4$ concentration of at least 0.5 ppmv above the outdoor ambient air $CH_4$ concentration (see S2 Appendix section 1). We chose this threshold based on prior experience that a 0.5 ppmv elevation in basement rooms relative to outdoors reliably had leaks and because this study was intended to resolve leaks which have greater significance as contributors to greenhouse gas emissions. The RCM and bag methods do not measure post-meter uncombusted methane emissions that may reach the outside of the building through other potential pathways such as flues, for which methods should also be developed. Both methods were tested in two different basements using controlled $CH_4$ releases at different $CH_4$ concentrations and different flow rates (see S6 Appendix). We then tested the application of the methods in a variety of basements. We also performed detailed indoor point-of-leak ("leak") surveys using a periodically calibrated Bascom-Turner Gas-Rover VG211 portable combustible gas indicator ("CGI") equipped with a rubber coned surface probe (SP-636) (see S9 Appendix section 3). Leak surveying was done across all accessible NG infrastructure and appliances on all floors of the buildings to detect any safety issues, learn about sources of indoor NG leaks and provide an indoor NG leaks amelioration plan to participants. Surveying covered interior service lines, meters, pipes, pipe joints, couplings and connectors, appliances such as furnaces, boilers, stoves, dryers and fireplaces.

## Recruitment of buildings

Participants' buildings were recruited through residential neighborhood associations, the Mothers Out Front network and the researchers' personal networks. In total, 23 buildings were available, spanning a variety of building styles, ages, floors, occupancy types, towns and neighborhoods. Participants were provided with a description of the study and phone-screened for eligibility. Buildings with whole-house HVAC systems in operation were not included as these interfered with the air flushing and circulation steps of the methods.

We selected 20 buildings for the study that were representative of the Boston area's housing stock and located in 8 different towns and neighborhoods (a map of locations and a table summarizing their characteristics can be found in S1 Appendix). Buildings studied included 14 single-family homes built between 1920 and 2005 (mean age 108 y, median age 102 y) covering a variety of architectural styles (3 colonials, 1 Victorian, 3 capes, 1 ranch, 2 bungalows, the rest 'conventional'). These buildings had 2–4 floors including the basement and basement volumes between 814–6787 ft$^3$ and 702–5707 ft$^3$ after adjusting for volume of objects therein. Buildings studied also included 6 multi-family homes built between 1870 and 1984 (mean age 106 y, median age 116 y) with styles covering one 3 story (3 families, 1 shared basement), 2 'conventional' (3 family and 2 family, both with shared basements), 2 duplex halves (both with divided

basements) and one 2 story (2 families, 1 shared basement). These buildings had 3–4 floors including the basement with basement volumes ranging between 2256–8549 ft$^3$ and 2169–8283 ft$^3$ after adjusting for the volume of objects therein. Across all 20 buildings, basements were on average approximately 75% subterranean and objects therein occupied on average 8% of their volumes. All but 3 had NG meters located inside the basement, and all but 3 had NG utility service pipes entering the basement from underground. All of the buildings had at least 1 NG furnace or boiler (27 in total) used for space heating or both space heating and domestic water heating; 16 buildings had NG powered water heaters; 11 buildings had NG dryers and 1 home had 1 NG fireplace. One of the buildings had a NG furnace in both the basement and the attic.

## Room chamber method ("RCM")

Exterior windows and doors were opened and basement air was flushed out and replaced with outdoor air using 3 to 4 adjustable-speed tilting-head 4960-CFM 51 cm floor fans (Pelonis model PFE50A4ABB) and 2 to 3 51 cm 1800 CFM box fans (Lasko model 3733). Air flushing continued until a flushed steady state was detected with the GasScouter where air CH$_4$ concentrations were approximately steady. Exterior windows and doors were then closed and the fans reoriented to provide maximum air CH$_4$ mixing and circulation in the basement. During this air CH$_4$ concentration rise phase, data was logged for as long as practicable (between 1.5 and 19 hours, after equipment set-up, air flushing and room volume measuring), and used for data modeling and emissions calculations. Fig 1 shows an example of the air CH$_4$ concentration across the three phases.

The basement volume was measured using a laser distance measurer (RockSeed Mileseey S2-50) and a tape measure. The basement was divided into spaces such as main rooms, stairwells, window bay cavities and crawl spaces. Their volumes were measured and summed to create a total 'empty' basement volume (V$_{empty}$). Volumes of objects (V$_{objects}$) in the basement

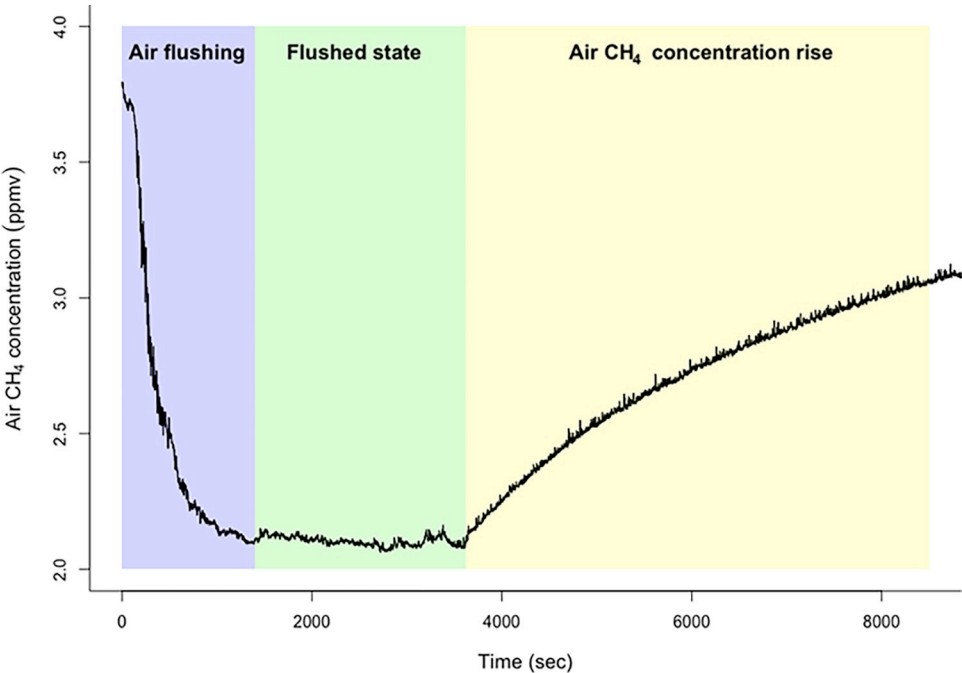

**Fig 1. Example of air CH$_4$ concentrations during each phase of the room chamber method (RCM).**

were measured and summed, including chimneys, water tanks, furnaces, boilers, washers and dryers, ceiling cross members and floor support beams or pillars, large furniture such as cupboards and sofas, heating ducts and plenums, and other miscellaneous objects. An adjusted basement volume was calculated as $V_{adj} = V_{empty} - V_{objects}$. We calculated the daily basement $CH_4$ emissions as follows (and provide $CH_4$ fluxes in cubic feet per day, rather than cubic meters per day as natural gas is measured in the United States in cubic feet; 1 cubic foot = 0.0283 cubic meters):

$$
\begin{aligned}
CH_4 \text{ emissions (ft}^3 \text{ day}^{-1}) = \\
dCH_4/dt \; (ppmv \; sec^{-1}) \\
x \; 60 \; sec \; x \; 60 \; min \; x \; 24 \; hr \; (day) \\
x \; V_{adj} \; (ft^3) \\
x \; 10^{-6}
\end{aligned}
\tag{Eq1}
$$

where $dCH_4/dt$ is the rate of change of air $CH_4$ concentration and $V_{adj}$ is the volume of the room chamber adjusted by removing the volume of objects therein. Two methods were compared for calculating $dCH_4/dt$. The first method ("simple slope") calculates the linear rate of change from two air $CH_4$ concentration measurements over time:

$$
dCH_4/dt = (CH_4 \text{ at } t_n - CH_4 \text{ at } t_0)/n
\tag{Eq2}
$$

where n is the timespan in seconds between two air $CH_4$ concentration measurements, $CH_4$ at $t_n$ is the air $CH_4$ concentration in the basement after n seconds, and $CH_4$ at $t_0$ is the air $CH_4$ concentration at the beginning of data logging (t = 0), after flushing and closing the basement (see S4 Appendix section 2 for an example). In the second method ("fitted tangent"), air $CH_4$ concentration over time in the room chamber is modeled with the equation:

$$
CH_4(t) = S - \alpha e^{-kt}
\tag{Eq3}
$$

where t is time, S is the ideal air $CH_4$ concentration steady state in the room chamber, k is a time constant, and $\alpha$ is some constant. The maximum rate of change of air $CH_4$ concentration is calculated using the first derivative of this equation at t = 0 (see S4 Appendix section 3 for example):

$$
dCH_4/dt = k\alpha
\tag{Eq4}
$$

The RCM method was tested against metered control $CH_4$ fluxes in 2 basements between June 12 2022 and August 2 2022. We tested 4 orders of magnitude of control fluxes ranging from 0.2 to 198 ft$^3$ day$^{-1}$. Although this study did not investigate control leak fluxes lower than 0.2 ft$^3$ day$^{-1}$, smaller leak fluxes could be investigated using this method. Using fitted tangent and simple slope calculation methods, RCM detected flux that was on average within 7% and 12% respectively of the control test fluxes (see S6 Appendix section 1). Having determined the accuracy of RCM, we performed the method between July 26 2022 and October 13 2022 in 16 basements and 1 attic (RCM was not performed in 3 basements that did not have an average air $CH_4$ concentration of at least 0.5 ppmv above the outdoor ambient air $CH_4$ concentration). The GasScouter was placed in an approximately central basement location and away from any gas appliances.

## Simplified room chamber method ("Bag method")

We explored the viability of a simpler version of RCM that could be used at scale using inexpensive equipment together with a centrally-located gas analyzer. Basement exterior windows and doors were opened and air was flushed out and replaced with outdoor air using 2 to 3 51 cm 1800-CFM box fans (Lasko model 3733). Basement air $CH_4$ concentrations were monitored with the GasScouter until a flushed state was observed in order to learn how long the air had to be flushed to reach a steady state. Air $CH_4$ concentration sampling was simplified by replacing the use of the GasScouter with an air sampling kit consisting of two 0.5 L Tedlar bags with polypropylene combo valves (Restek 22049) and a gas sampling bulb (Heathrow Scientific 56HV89) used to evacuate and fill the bags. A first air $CH_4$ concentration sample was taken in one bag after basement air flushing was completed and before basement doors and windows were closed. Basement exterior doors and windows were then closed and fans reoriented to provide air $CH_4$ mixing and circulation in the basement. After a recorded amount of time later (typically 600 s), a second air sample was taken using the second bag (details on using the air sampling kit can be found in S5 Appendix section 1). Bagged air $CH_4$ concentrations were later analyzed using the GasScouter or a Picarro G2311-f cavity ring-down spectrometer which were tested periodically with test gasses (see S9 Appendix section 2). Measurement of $V_{empty}$ was simplified by omitting small and harder-to-measure sub-volumes such as basement stairway areas above the basement ceilings, window bays or small built-in shelf or storage spaces. A standard adjustment factor was then applied to $V_{empty}$ to account for basement contents, resulting in a final simplified volume, $V_{sim} = V_{empty} \times f$, where $f$ was derived by taking the average volume consumed by objects in basements measured across this study ($f = 0.92$) (see S4 Appendix section 1). We calculated the daily basement $CH_4$ emissions using Eq 1, replacing $V_{adj}$ with $V_{sim}$ and calculating $dCH_4/dt$ using simple slope (Eq 2) with $CH_4$ at $t_0$ and $CH_4$ at $t_n$ taken from the first and second bagged air samples respectively.

The bag method was tested against metered control $CH_4$ fluxes in 1 basement between June 12 2022 and October 10 2022 using test fluxes ranging from 0.19 to 1.03 $ft^3$ $day^{-1}$ (S6 Appendix section 2). Having determined the accuracy of the bag method, we performed the method in 14 basements (11 immediately prior to RCM and in 3 basements on different days to RCM) between September 3 2022 and October 13 2022.

# Results and discussion

## Room chamber method (RCM)

Air $CH_4$ concentrations during flushing reduced exponentially over time (see S3 Appendix). Total basement air flushing time (including the bag method flushing duration) ranged between 25–120 min (mean 68, median 60 min, n = 16) and post-flushed basement air $CH_4$ concentrations relative to outdoors had a mean and median of 1% higher relative to the outdoors. We suggest that this small difference may be due to active NG leakage and slower air/$CH_4$ exchange rates from stored object volumes (e.g. storage boxes, furniture, appliances). During RCM we collected 1.5–19 h (mean 7 h, median 5.3 h, n = 18) of data with the GasScouter. During the post-flush rise phase at three experiment locations, we observed spikes in air $CH_4$ concentration possibly due to unburnt gas entering the basement from the cycling on/off of domestic water heating NG furnaces or boilers [23] (see S4 Appendix section 6). We also observed examples of air $CH_4$ concentration fluctuating after initial concentration rises (see S4 Appendix section 7). However, neither of these issues adversely affected our ability to model the data for our purposes and precision sought. During the rise phase at one experiment location, we observed high frequency large variations in air $CH_4$ concentration readings due to

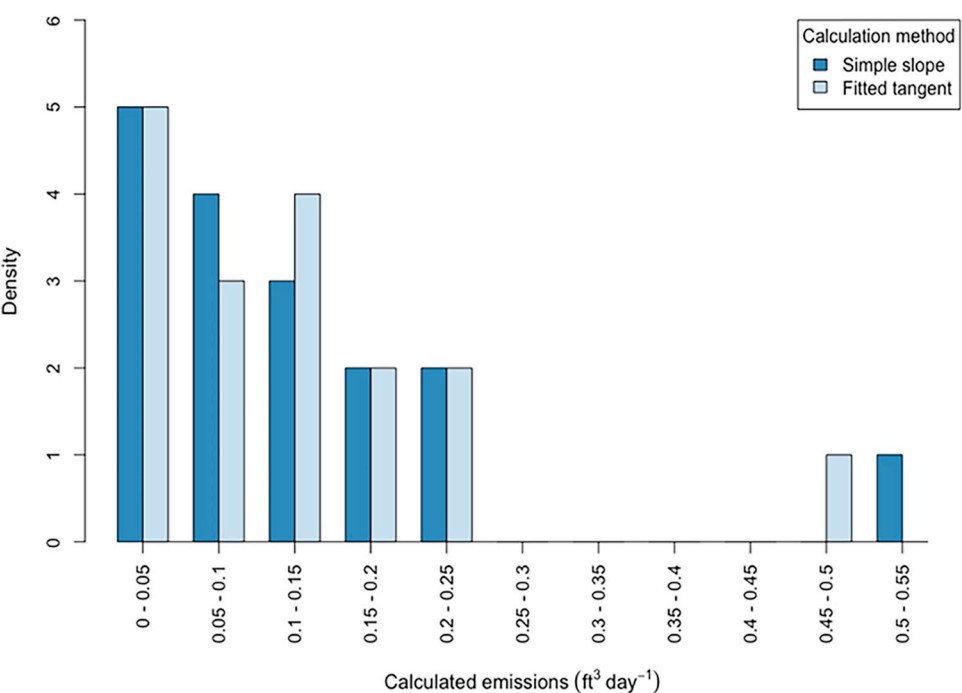

**Fig 2. Histogram of CH$_4$ calculated emissions from room chamber method (RCM), comparing results from using the simple slope and fitted tangent emissions calculation methods.**

poor air circulation. This was addressed by repeating the experiment after closing off spaces and sub-rooms in the basement where it was difficult to circulate air and where there was no NG infrastructure (see S4 Appendix section 8). We also explored how indoor air CH$_4$ concentration may vary with indoor and outdoor atmospheric pressure by performing a multi-day experiment in one basement in which all NG appliances in the basement (and entire building) were turned off. We found no relationship between pressure and CH4 concentration (see S4 Appendix section 9). It is possible that fluctuations in air CH$_4$ concentrations in this scenario could be due to pressure fluctuations in the NG distribution system. After taking fluctuations and spikes into account, we selected as much exponential rise phase data as possible (between 0.3–6.6 h, mean 3.3 h, median 3.2 h) for use in modeling in 16 basements and 1 attic room (see S10 Appendix for details of each experiment). Calculated emissions ranged between 0.02–0.51 ft$^3$ day$^{-1}$ and are shown for both calculation methods as histograms in Fig 2. Mean daily emissions were 0.13 and 0.12 ft$^3$ day$^{-1}$ using simple slope and fitted tangent methods respectively, and median daily emissions were 0.09 and 0.1 ft$^3$ day$^{-1}$ using simple slope and fitted tangent methods respectively (n = 17).

Our calculated emissions from inside basement rooms were comparable in magnitude to findings in the Fischer et al study (Fischer et al., 2018) which measured mean and median whole-house emissions of 0.10 ft$^3$ day$^{-1}$ and 0.23 ft$^3$ day$^{-1}$ (2.1 and 4.6 g CH$_4$ day$^{-1}$) respectively (n = 75). We recognize that the different methods likely probe different flow regimes i.e. pressure driven mass flow in the Fischer et al study versus molecular diffusion dominated-flow in our study (as discussed in Bain et al., 2005). We also recognize that our measurements were not for the whole house, and were also performed on a smaller number and wider variety of buildings in Massachusetts. As an additional point of comparison, our calculated emissions are less than 1% of the average daily emissions of 89 ft$^3$ day$^{-1}$ from 79 distribution system leaks studied by Magavi in 2018 in the greater Boston area [24]. The largest calculated daily

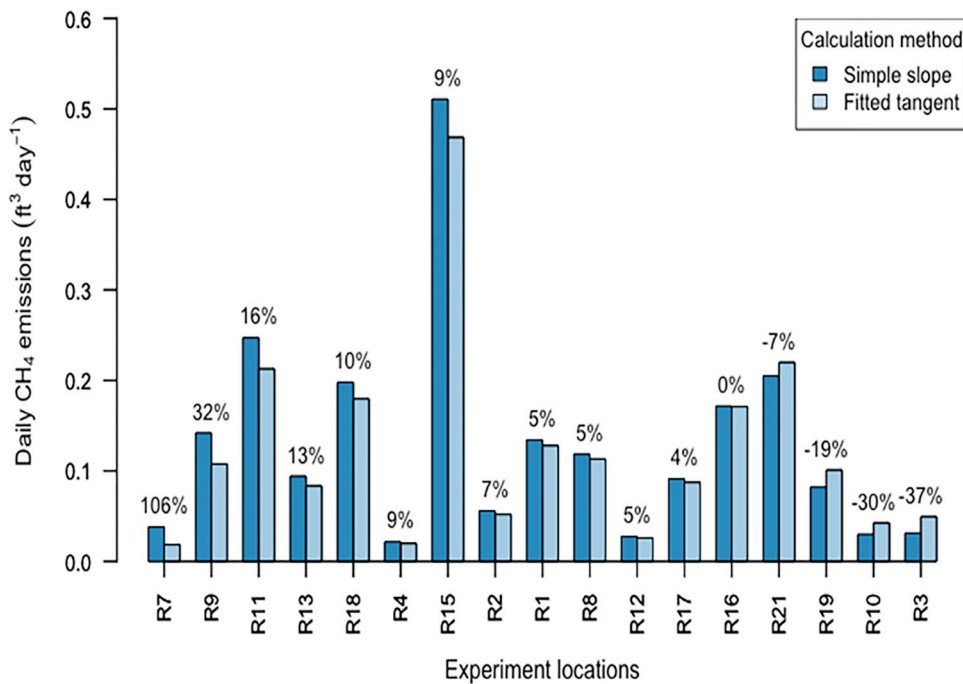

**Fig 3. Daily CH$_4$ emissions calculated using room chamber method (RCM) ranged between 0.02–0.51 ft$^3$ day$^{-1}$.**
The relative difference between emissions from the two RCM calculation methods is shown as a percentage above each pair of bars. The graph is ordered from left to right in descending order of the relative percentage difference.

emissions from one of the basements, regardless of calculation method, was responsible for 23% of total emissions across all experiment locations (4 standard deviations above the mean), suggesting there might be super-emitting basements. Fig 3 shows a comparison of emissions using the two calculation methods which correlated well ($R^2$ = 0.983, p < 0.001). Emission calculations using simple slope were between 37% less to 106% more when compared relatively to emissions calculated using fitted tangent, with a mean relative difference of 8% and median relative difference of 5%.

When calculating daily emissions with either calculation method, as the timespan used in calculations increased, calculated emissions decreased. We analyzed a 30 minute control test by varying the timespan from 60 to 1680 s in 60 s increments and found emissions calculated using simple slope diverged from 102% to 93% of the actual test flux. Similarly, when using fitted tangent, varying the timespan from 100 s to 1700 s in 60 s increments, calculated emissions diverged from 96% to 92% of the actual test flux (see S4 Appendix section 4). For simple slope calculations we chose to use a timespan of 600 seconds. Differences between calculated emissions and actual emissions could in part be explained by error in volume measurements.

## Simplified room chamber method ("Bag method")

Flush duration and post-flush air CH$_4$ concentration was measured in the 11 experiment locations performed immediately prior to RCM. As with RCM, air CH$_4$ concentrations during flushing reduced exponentially over time (see S3 Appendix). Flush time ranged between 35–60 min with both a mean and median of 48 min. Post-flushed basement air CH$_4$ concentrations relative to outdoors were slightly higher than RCM with a mean 7% and median 6% higher relatively. Daily emissions calculated using simplified volumes (V$_{sim}$) and the more accurate adjusted volumes (V$_{adj}$) are shown as histograms in Fig 4 and ranged between 0.02–0.55 ft$^3$

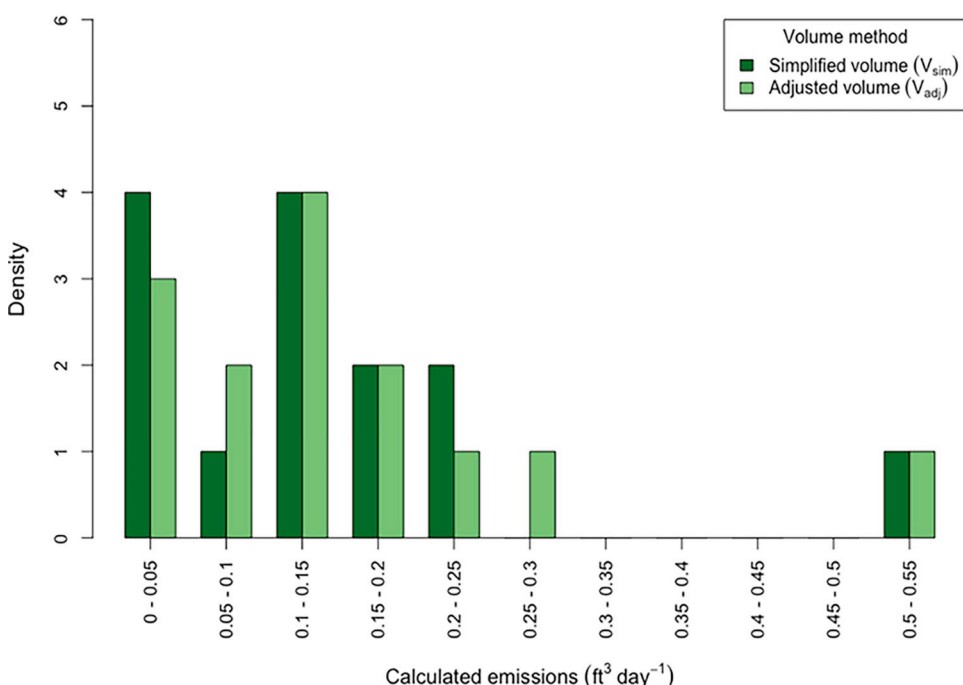

**Fig 4. Histogram of CH$_4$ calculated emissions from the bag method, comparing results of using the simplified room volume (V$_{sim}$) and the more accurate room volume (V$_{adj}$).**

day$^{-1}$. Mean daily emissions were 0.14 and 0.15 ft$^3$ day$^{-1}$ using V$_{sim}$ and V$_{adj}$ respectively, and median daily emissions were 0.14 and 0.13 ft$^3$ day$^{-1}$ using V$_{sim}$ and V$_{adj}$ respectively (n = 14).

Emissions calculated using V$_{sim}$ and V$_{adj}$ were on average lower by 15% and 14% respectively than metered control tests but still within our sought after level of precision. Simplified volumes differed from 8% less to 11% greater than adjusted volumes and on average simplified volumes were 3% less than adjusted volumes (n = 15). Emissions calculations using V$_{sim}$ and V$_{adj}$ correlated well (R$^2$ = 0.997, p < 0.001) and V$_{sim}$ calculations when compared relatively to V$_{adj}$ differed between 9% less to 11% more in result, with an average difference of 3% less and median difference of 4% less (Fig 5).

Fig 6 shows the relative differences between the bag method calculations using V$_{sim}$ and RCM calculations using simple slope. These correlated well (R$^2$ = 0.946, p < 0.001) and ranged from 51% less to 118% greater than RCM emissions, with a mean 26% greater and a median 27% greater (n = 14). These differences may likely be due to the less thorough air flushing used in the bag method, and secondarily the slight variation in volume measurements. Also, the relative difference between the bag method (using V$_{sim}$) and RCM simple slope calculated emissions appeared to reduce as emissions calculated with RCM increases (see S5 Appendix section 2). Due to its lower sampling rate, this method's results could be more susceptible to air CH$_4$ concentration spikes from NG appliance cycling. For example, if the second sample were taken nearer in time or space to a spike, the calculated flux might appear larger. Sampling protocol could be further improved, for example by performing the air flush/sample steps multiple times.

## Ambient air CH$_4$ concentration measurements

Across all single-family homes, ambient air CH$_4$ concentrations were on average highest in the basement (3.8 ppmv, n = 15). Across all multi-family homes, on average the highest reading

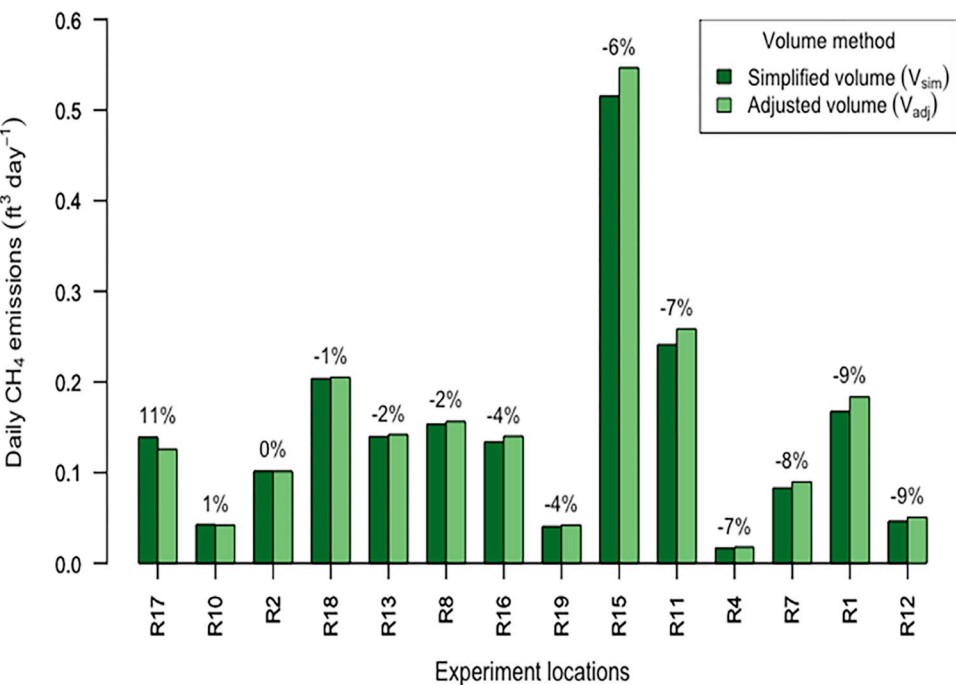

**Fig 5. Daily CH$_4$ emissions calculated using the bag method ranged between 0.02–0.55 ft$^3$ day$^{-1}$.** The relative difference between emissions from the bag method calculations using simplified volume (V$_{sim}$) and the more accurate volume (V$_{adj}$) is shown as a percentage above each pair of bars. The graph is ordered from left to right in descending order of the relative percentage difference.

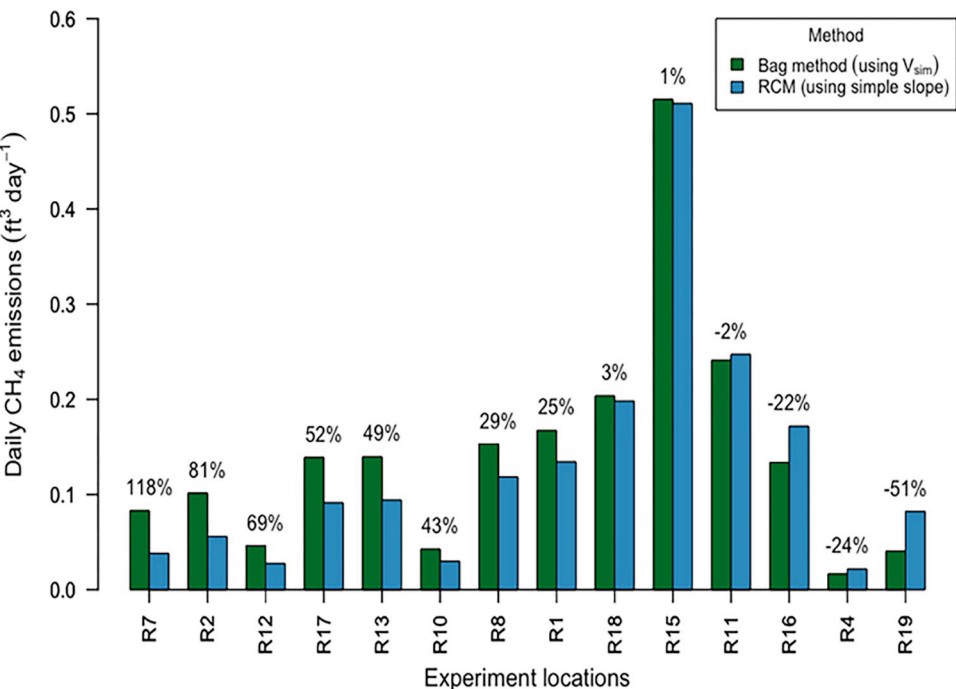

**Fig 6. Comparison of calculated emissions using the bag method and room chamber method (RCM).** The relative difference in calculated emissions between the two methods is shown as a percentage above each pair of bars. The graph is ordered from left to right in descending order by relative percentage difference. (Note: The bag method was performed on different days to RCM for experiment locations R12, R13 and R16).

was on floor 3 (16.6 ppmv, n = 5) and was due to one building having a floor 3 ambient air $CH_4$ concentration of 16.6 ppmv due to an extinguished gas stove pilot light (no other leaks were found through indoor leak surveying on this floor). Air $CH_4$ concentrations in these basements were on average 3.5 ppmv (n = 6) (see S2 Appendix section 2). We found a correlation between unflushed ambient basement air $CH_4$ concentrations and RCM daily emissions calculated using fitted tangent, the most accurate emissions method and calculation relative to control tests ($R^2$ = 0.39, p = 0.0075, n = 17, see S4 Appendix section 5). The basement's air exchange rate could account for much of the unexplained variation in this result.

## Leak measurements

Of all the buildings studied, 18 out of 20 (90%) had at least 1 indoor gas leak detected during surveying. The majority of found leaks were in the basement compared to other floors, and were distributed approximately evenly between single and multi-family home types. Leaks were found both before and after the NG meter (15 and 58 leaks respectively). Leaks after the NG meter occurred slightly more frequently in multi-family homes (59%) and leaks before the NG meter which occurred more frequently in single-family homes (87%). The majority (83%) of the leaks were found at pipe joints. No leaks were found on meters. More details about leaks surveyed can be found in S7 Appendix. Also, building age didn't appear to correlate with the number of leaks, their sizes or calculated daily emissions (see S8 Appendix).

## Conclusions

In general both RCM and bag method can be used to measure indoor uncombusted $CH_4$ emissions in any building room where the air in the space can be flushed sufficiently and air can be thoroughly circulated. While the bag method can potentially be affected by air $CH_4$ concentration spikes from NG appliance cycling, when compared to RCM it was faster to perform, required simpler on-site equipment and room volume measurements and used simple emissions calculations resulting in only slightly less accuracy when compared to metered control tests results. Room flushing and air sampling could be performed by trained community-based scientists and air $CH_4$ concentration bagged air samples can easily be processed at centralized locations for gas analysis. In addition, the bag method could potentially be used as a widely-adoptable screening tool to detect post-meter super-emitting gas leaks that are likely associated with long-tailed leak distributions. Super-emitting leaks have characterized all other parts of the NG process chain and characterizing long-tailed statistical distributions benefits from large data sets. Such a scalable remote sampling and centralized analysis approach could play an important role in dramatically increasing sampling rates both seasonally and spatially, and could lead to improvements in indoor $CH_4$ emissions research and bottom-up post-meter methane emissions inventories. In the next phase of research we intend to actively pursue these applications.

## Supporting information

**S1 Appendix. Selected building locations and characteristics.**
(PDF)

**S2 Appendix. Ambient $CH_4$ air concentration measurements.**
(PDF)

**S3 Appendix. Air $CH_4$ concentrations over time during air flushing phase.**
(PDF)

**S4 Appendix. Room chamber method (RCM).**
(PDF)

**S5 Appendix. Simplified room chamber method ("Bag method").**
(PDF)

**S6 Appendix. Testing methods with metered CH₄ releases.**
(PDF)

**S7 Appendix. Leaks.**
(PDF)

**S8 Appendix. Building age relationships.**
(PDF)

**S9 Appendix. Instrument quality control tests.**
(PDF)

**S10 Appendix. Individual RCM experiment data and modeling plots.**
(PDF)

**S1 Data.**
(ZIP)

## Acknowledgments

The authors would like to warmly thank the study participants and Sarah Lerman-Sinkoff, Dr Marcos Luna, Audrey Schulman and Zeyneb Magavi for thoughtful reviews and suggestions.

## Author Contributions

**Conceptualization:** Dominic Nicholas, Robert Ackley, Nathan G. Phillips.

**Data curation:** Dominic Nicholas.

**Formal analysis:** Dominic Nicholas.

**Investigation:** Dominic Nicholas, Robert Ackley, Nathan G. Phillips.

**Methodology:** Dominic Nicholas, Robert Ackley, Nathan G. Phillips.

**Project administration:** Dominic Nicholas.

**Resources:** Dominic Nicholas, Robert Ackley, Nathan G. Phillips.

**Software:** Dominic Nicholas.

**Supervision:** Dominic Nicholas, Robert Ackley, Nathan G. Phillips.

**Validation:** Dominic Nicholas, Robert Ackley, Nathan G. Phillips.

**Visualization:** Dominic Nicholas.

**Writing – original draft:** Dominic Nicholas.

**Writing – review & editing:** Dominic Nicholas, Robert Ackley, Nathan G. Phillips.

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
