## [Decision Letter · Decision Letter 0]

26 Sep 2023

PONE-D-23-18731A Simple Method To Measure Methane Emissions From Indoor Gas LeaksPLOS ONE

Dear Dr. Nicholas,

Thank you for submitting your manuscript to PLOS ONE. After careful consideration, we feel that it has merit but does not fully meet PLOS ONE’s publication criteria as it currently stands. Therefore, we invite you to submit a revised version of the manuscript that addresses the points raised during the review process.

We look forward to receiving your revised manuscript.

Kind regards,

Yanping Yuan

Academic Editor

PLOS ONE

Journal Requirements:

4. We note that Supporting Figure 1 in your submission contain map images which may be copyrighted. All PLOS content is published under the Creative Commons Attribution License (CC BY 4.0), which means that the manuscript, images, and Supporting Information files will be freely available online, and any third party is permitted to access, download, copy, distribute, and use these materials in any way, even commercially, with proper attribution. For these reasons, we cannot publish previously copyrighted maps or satellite images created using proprietary data, such as Google software (Google Maps, Street View, and Earth). For more information, see our copyright guidelines: http://journals.plos.org/plosone/s/licenses-and-copyright.

(1) You may seek permission from the original copyright holder of Supporting Figure 1 to publish the content specifically under the CC BY 4.0 license.  

5. We note that Supporting Figures R3 & R4, R9, R10, R15, and R21 in your submission contain copyrighted images. All PLOS content is published under the Creative Commons Attribution License (CC BY 4.0), which means that the manuscript, images, and Supporting Information files will be freely available online, and any third party is permitted to access, download, copy, distribute, and use these materials in any way, even commercially, with proper attribution. For more information, see our copyright guidelines: http://journals.plos.org/plosone/s/licenses-and-copyright.

(1) You may seek permission from the original copyright holder of Supporting Figures R3 & R4, R9, R10, R15, and R21 to publish the content specifically under the CC BY 4.0 license.

6.. We are unable to open your Supporting Information file [rawdata.gzip]. Please kindly revise as necessary and re-upload.

Reviewers' comments:

Reviewer's Responses to Questions

**Comments to the Author**

1. Is the manuscript technically sound, and do the data support the conclusions?

Reviewer #1: Yes

Reviewer #2: Yes

2. Has the statistical analysis been performed appropriately and rigorously? 

Reviewer #1: Yes

Reviewer #2: Yes

3. Have the authors made all data underlying the findings in their manuscript fully available?

Reviewer #1: Yes

Reviewer #2: Yes

4. Is the manuscript presented in an intelligible fashion and written in standard English?

Reviewer #1: Yes

Reviewer #2: Yes

5. Review Comments to the Author

Reviewer #1: The article is well written and organized, and the study was very thoroughly conducted. This method should provide a solution to the problem of quickly and inexpensively obtaining data for building methane emissions. My suggestions for revision are minor.

Line 152: What is meant by "as long as possible"? Be specific.

Line 205: What was the rationale for choosing 0.5 ppmv?

What are the detection limits of both approaches? This should be included.

Was there any activity in the buildings during the tests, for example, gas dryers or furnaces running/not running? (And how might this impact results.) The cycling of water heaters was mentioned, and since it may have had an impact on results this issue seems important to address.

Line 252 and again on line 331: What is "precision sought"? (See detection limit comment above; if a detection limit was given you would be able to define precision sought.

Line 261: "After taking fluctuations and spikes into account": does this mean that these data points were considered outliers and excluded from the analysis? What was the decision making process?

Also line 261: Do we know how much pressure can fluctuate in the NG distribution system? I would guess these fluctuations would have to be small.

Line 308: I find it interesting that emissions changed linearly with basement volume. When you consider the potential sources, some are likely the same (one gas dryer, one water heater, etc.) regardless of a big or small basement. Comments on this?

Reviewer #2: The contribution proposes a new and simple method to measure methane emissions from indoor sources, which is an insufficiently considered. Better assessing these emissions could reduce the gap between top-down and bottom-up estimates. The experiments have been properly conducted, and the conclusions clearly reflect the obtained results.

Mathematical and statistical data processing is appropriate. All data underlying the findings is fully available. The manuscript is written in a clear manner, with a high quality of English language.

6. PLOS authors have the option to publish the peer review history of their article (what does this mean?). If published, this will include your full peer review and any attached files.

Reviewer #1: No

Reviewer #2: No

---

## [Author Response · Author response to Decision Letter 0]

24 Oct 2023

(The content below can be found with correct formatting in the "Response to Reviewers.docx" file submitted - thanks)

October 24, 2023

Dear Editor Yuan and reviewers, 

Thank you for your valuable comments and suggestions. They have significantly strengthened this manuscript. Below we have provided responses and updates as per the journal requirements in items 1-7 below. We have also responded in detail to each comment and question of the reviewers and specified where the manuscript and supporting information have been edited. We have uploaded change-tracking and clean versions of the manuscript and SI for your review.

Dominic Nicholas, Bob Ackley, Nathan Phillips 

Response to Reviewers

Author’s response:

Style requirements checked and minor updates listed below: 

Manuscript:

Level 1 headings : font size increased from 17 to 18

Level 2 headings : font size increased from 11 to 16

Changed “Figure” to “Fig” and bolded first sentence of each figure’s caption

Double-spacing applied to entire manuscript

Corresponding author format revised by adding *

Supporting information:

Corresponding author format revised by adding *

See 3 below for other updates

Author’s response:

The code written for this paper is publicly available through the github repository https://github.com/dominicnicholas/research and shared using a GNU General Public License v3.0.

Author’s response:

Individual PDF files of each section of the Supporting Information have been created and captions are included at the end of the complete and updated Supporting Information document. For manuscript SI references of the format S#.# (e.g. S9.1), please hyperlink to the main section’s PDF (e.g. S9.1 should hyperlink to S9.pdf). All other manuscript SI references are of the form S# and should hyperlink to S#.pdf (e.g. S1 should hyperlink to S1.pdf). Supporting Information captions and file associations are as follows:

SI Caption

Associated SI PDF file

S1. Selected Building Locations and Characteristics.

S1_File.pdf

S2. Ambient CH4 air concentration measurements.

S2_File.pdf

S3. Air CH4 concentrations over time during air flushing phase.

S3_File.pdf

S4. Room Chamber Method (RCM).

S4_File.pdf

S5. Simplified Room Chamber Method (“Bag method”).

S5_File.pdf

S6. Testing methods with metered CH4 releases.

S6_File.pdf

S7. Leaks.

S7_File.pdf

S8. Building age relationships.

S8_File.pdf

S9. Instrument Quality Control Tests.

S9_File.pdf

S10. Individual RCM experiment data and modeling plots.

S10_File.pdf

4. We note that Supporting Figure 1 in your submission contain map images which may be copyrighted. All PLOS content is published under the Creative Commons Attribution License (CC BY 4.0), which means that the manuscript, images, and Supporting Information files will be freely available online, and any third party is permitted to access, download, copy, distribute, and use these materials in any way, even commercially, with proper attribution. For these reasons, we cannot publish previously copyrighted maps or satellite images created using proprietary data, such as Google software (Google Maps, Street View, and Earth). For more information, see our copyright guidelines: http://journals.plos.org/plosone/s/licenses-and-copyright.

(1) You may seek permission from the original copyright holder of Supporting Figure 1 to publish the content specifically under the CC BY 4.0 license. 

Author’s response:

The supporting figure 1 map has been replaced with a public domain OpenStreetMap (OSM) static image of less than 100 features. Whilst not strictly required by OSM for a map with this number of features, we’ve included an embedded attribution (OSM attribution guidelines are defined in https://osmfoundation.org/wiki/Licence/Attribution_Guidelines#Static_images)

5. We note that Supporting Figures R3 & R4, R9, R10, R15, and R21 in your submission contain copyrighted images. All PLOS content is published under the Creative Commons Attribution License (CC BY 4.0), which means that the manuscript, images, and Supporting Information files will be freely available online, and any third party is permitted to access, download, copy, distribute, and use these materials in any way, even commercially, with proper attribution. For more information, see our copyright guidelines: http://journals.plos.org/plosone/s/licenses-and-copyright.

(1) You may seek permission from the original copyright holder of Supporting Figures R3 & R4, R9, R10, R15, and R21 to publish the content specifically under the CC BY 4.0 license.

Author’s response:

Supporting figures R3, R4, R9, R10, R15, and R21 are all original figures generated with the author’s original R code and derived from data from this research. These figures are not copyrighted.

6.. We are unable to open your Supporting Information file [rawdata.gzip]. Please kindly revise as necessary and re-upload.

Author’s response:

The two originally submitted data files “research data and code.zip” and “rawdata.gzip” have been reorganized and repackaged into one ‘data and code.zip’ 73.1 MB ZIP archive file.

Author’s response:

References were reviewed and hyperlinks were corrected and improved.

Reviewers' comments:

Reviewer's Responses to Questions

Comments to the Author

1. Is the manuscript technically sound, and do the data support the conclusions?

Reviewer #1: Yes

Reviewer #2: Yes

2. Has the statistical analysis been performed appropriately and rigorously?

Reviewer #1: Yes

Reviewer #2: Yes

3. Have the authors made all data underlying the findings in their manuscript fully available?

Reviewer #1: Yes

Reviewer #2: Yes

4. Is the manuscript presented in an intelligible fashion and written in standard English?

Reviewer #1: Yes

Reviewer #2: Yes

5. Review Comments to the Author

Reviewer #1: The article is well written and organized, and the study was very thoroughly conducted. This method should provide a solution to the problem of quickly and inexpensively obtaining data for building methane emissions. My suggestions for revision are minor.

Line 152: What is meant by "as long as possible"? Be specific.

Author’s response

Thank you for this good suggestion. We have updated the manuscript as follows with new content in bold, and removed words crossed out (lines 158-159 of revised manuscript) :

“During this air CH4 concentration rise phase, data was logged for as long as possible practicable (between 1.5 and 19 hours, after equipment set-up, air flushing and room volume measuring), and used for data modeling and emissions calculations. Figure 1 shows an example of the air CH4 concentration across the three phases.”

Line 205: What was the rationale for choosing 0.5 ppmv?

Author’s response:

Thank you for your response. We have clarified this in the first mention of this threshold in the Measurements Overview section as follows (lines 106-109 of revised manuscript): 

“We chose this threshold based on prior experience that a 0.5 ppmv elevation in basement rooms relative to outdoors reliably had leaks and because this study was intended to resolve leaks which have greater significance as contributors to greenhouse gas emissions.”

What are the detection limits of both approaches? This should be included.

Author’s response:

Thank you again. Although the detection limits of the instruments used here are listed in SI Tables 5 and 6, we did not test the lower detection limit of either RCM or Bag methods. We are confident in the precision and accuracy of the methods over the range of control fluxes verified in this study (S6). We suggest that researchers who are interested in resolving smaller leak fluxes can use this approach with appropriate control tests. To help clarify, we have edited the manuscript Materials and Methods Room Chamber Method (“RCM”) section (lines 211-212 of revised manuscript) to include the following : 

“Although this study did not investigate control leak fluxes lower than 0.2 ft3 day-1, smaller leak fluxes could be investigated using this method.”

Was there any activity in the buildings during the tests, for example, gas dryers or furnaces running/not running? (And how might this impact results.) The cycling of water heaters was mentioned, and since it may have had an impact on results this issue seems important to address.

Author’s response:

No NG dryers or NG central heating systems were in use during the experiments. In some buildings, domestic water heating NG furnaces or boilers were observed to be cycling on/off (as we note on lines 247-250). Different times of year might result in different frequency of spikes (eg winter and furnace cycling) and it will be interesting to explore. We’ve improved the conclusion section by changing the word ‘temporally’ to ‘seasonally’ (line 429 of revised manuscript).

With RCM, methane concentration spikes due to cycling of appliances were clearly visible by inspection of results. We tested the effect of these intermittent short duration spikes by including or excluding them in RCM modeling, and have added the results in S10 to the affected experiments R10, R15 and R16.Calculated fluxes were all within the same order of magnitude leak rate (0.04 vs 0.19, 0.47 vs 0.49 and 0.17 vs 0.16 ft3 CH4 day-1).

Bag Method results could be impacted by fuel slip spikes and the sampling protocol could be further developed to increase this method’s resilience. We have added the following sentence to the last paragraph of the Results and Discussion -> Simplified Room Chamber Method (“Bag method”) (lines 374-378 of revised manuscript) :

“Due to its lower sampling rate, this method’s results could be more susceptible to air CH4 concentration spikes from NG appliance cycling. For example, if the second sample were taken nearer in time or space to a spike, the calculated flux might appear larger. Sampling protocol could be further improved, for example by performing the air flush/sample steps multiple times.”

We have also changed the word ‘theoretically’ to ‘potentially’ in the Conclusions section (line 417 of revised manuscript).

Line 252 and again on line 331: What is "precision sought"? (See detection limit comment above; if a detection limit was given you would be able to define precision sought.

Author’s response:

We wanted to develop a method that provides us with an order-of-magnitude of emissions (0.1, 1, 10, 100 ft3 CH4 day-1). We did not seek a precision or accuracy less than 0.2 ft3 CH4 day-1, because we are not interested in detecting smaller leak fluxes with this method. We have updated the Abstract (lines 21-24 of revised manuscript) as follows:

‘We develop and test a simple and widely deployable closed chamber method that can be used for quantifying indoor methane emissions at a with an order-of-magnitude precision which allows for screening of indoor large volume (“super-emitting”) leaks.’

Line 261: "After taking fluctuations and spikes into account": does this mean that these data points were considered outliers and excluded from the analysis? What was the decision making process?

Author’s response: 

We modeled RCM fluxes with and without spikes and fluctuations being included and results remained within the same order of magnitude, i.e. in our sought after level of precision. We discussed spikes in our response to your earlier question regarding activity of NG appliances. 

In some experiments (R2, R9, R13, R16, R18, R19, R21), we observed fluctuations after the initial rise phase. For these, we had enough data to model with RCM that didn’t include fluctuations.

In experiments where we observed fluctuations affecting the rise phase (R1, R3, R4, R11), we tested the effect of these fluctuations by including or excluding them in RCM modeling, and have added the results in S10 to the affected experiments. In these comparison sets, calculated fluxes were all within the same order of magnitude leak rate (0.13 vs 0.13, 0.05 vs 0.02, 0.02 vs 0.03 and 0.21 vs 0.17 ft3 CH4 day-1).

Our decision making process also included prioritizing the use of a data range that was observed to be the initial rise phase prior to a plateau (or steady state). 

Also line 261: Do we know how much pressure can fluctuate in the NG distribution system? I would guess these fluctuations would have to be small.

Author’s response:

We did not measure supply side or post-meter pressure fluctuations. There is some evidence that pressure does fluctuate on the NG distribution system based on demand across the year. Also, according to at least one utility spokesperson : “most of the year pressures do not fluctuate more than about 20% on many of our different systems, but during winter, with very cold weather and high gas consumption, the pressure ranges can be considerably more than 20%.”. Seasonal NG distribution system pressure fluctuations may be significant, but that would not have been a concern within the minute-to-hour time scales studied here. Our future work explores measuring distribution system pressure with smart gas meters.

Line 308: I find it interesting that emissions changed linearly with basement volume. When you consider the potential sources, some are likely the same (one gas dryer, one water heater, etc.) regardless of a big or small basement. Comments on this?

Author’s response:

Thank you for your point. After reviewing this line, we felt it was potentially confusing and have removed it (from equation 1 it should be clear that varying the room volume will result in a directly proportional change in the flux result). The room air CH4 concentration measurements do not use room volume.

Thank you Reviewer #1 for taking the time and effort to provide your thoughtful and helpful comments, questions and input. It is very much appreciated. 

Reviewer #2: The contribution proposes a new and simple method to measure methane emissions from indoor sources, which is an insufficiently considered. Better assessing these emissions could reduce the gap between top-down and bottom-up estimates. The experiments have been properly conducted, and the conclusions clearly reflect the obtained results.

Mathematical and statistical data processing is appropriate. All data underlying the findings is fully available. The manuscript is written in a clear manner, with a high quality of English language.

Author’s response:

Thank you Reviewer #2 for your review and generous comments.

6. PLOS authors have the option to publish the peer review history of their article (what does this mean?). If published, this will include your full peer review and any attached files.

Do you want your identity to be public for this peer review? For information about this choice, including consent withdrawal, please see our Privacy Policy.

Reviewer #1: No

Reviewer #2: No

---

## [Decision Letter · Decision Letter 1]

15 Nov 2023

A Simple Method To Measure Methane Emissions From Indoor Gas Leaks

PONE-D-23-18731R1

Dear Dr. Nicholas,

We’re pleased to inform you that your manuscript has been judged scientifically suitable for publication and will be formally accepted for publication once it meets all outstanding technical requirements.

Kind regards,

Yanping Yuan

Academic Editor

PLOS ONE

Additional Editor Comments (optional):

Reviewers' comments:

Reviewer's Responses to Questions

**Comments to the Author**

1. If the authors have adequately addressed your comments raised in a previous round of review and you feel that this manuscript is now acceptable for publication, you may indicate that here to bypass the “Comments to the Author” section, enter your conflict of interest statement in the “Confidential to Editor” section, and submit your "Accept" recommendation.

Reviewer #1: All comments have been addressed

2. Is the manuscript technically sound, and do the data support the conclusions?

Reviewer #1: Yes

3. Has the statistical analysis been performed appropriately and rigorously? 

Reviewer #1: Yes

4. Have the authors made all data underlying the findings in their manuscript fully available?

Reviewer #1: Yes

5. Is the manuscript presented in an intelligible fashion and written in standard English?

Reviewer #1: Yes

6. Review Comments to the Author

Reviewer #1: (No Response)

7. PLOS authors have the option to publish the peer review history of their article (what does this mean?). If published, this will include your full peer review and any attached files.

Reviewer #1: No

---

## [Editor Report · Acceptance letter]

21 Nov 2023

PONE-D-23-18731R1 

A Simple Method To Measure Methane Emissions From Indoor Gas Leaks 

Dear Dr. Nicholas:

I'm pleased to inform you that your manuscript has been deemed suitable for publication in PLOS ONE. Congratulations! Your manuscript is now with our production department. 

Kind regards, 

on behalf of

Prof. Yanping Yuan 

Academic Editor

PLOS ONE